# Buckwheat and Cardiometabolic Health: A Systematic Review and Meta-Analysis

**DOI:** 10.3390/jpm12121940

**Published:** 2022-11-22

**Authors:** Erand Llanaj, Noushin Sadat Ahanchi, Helga Dizdari, Petek Eylul Taneri, Christa D. Niehot, Faina Wehrli, Farnaz Khatami, Hamidreza Raeisi-Dehkordi, Lum Kastrati, Arjola Bano, Marija Glisic, Taulant Muka

**Affiliations:** 1Department of Molecular Epidemiology, German Institute of Human Nutrition Potsdam-Rehbrücke, 14558 Nuthetal, Germany; 2Epistudia, 3012 Bern, Switzerland; 3ELKH-DE Public Health Research Group of the Hungarian Academy of Sciences, Department of Public Health and Epidemiology, Faculty of Medicine, University of Debrecen, Kassai út 26, 4028 Debrecen, Hungary; 4Institute of Social and Preventive Medicine (ISPM), University of Bern, 3012 Bern, Switzerland; 5School of Nursing and Midwifery, National University of Ireland Galway, H91 TK33 Galway, Ireland; 6HRB-Trials Methodology Research Network, H91 TK33 Galway, Ireland; 7Literature Searches Support, 3311 Dordrecht, The Netherlands; 8Department of Community Medicine, School of Medicine, Tehran University of Medical Sciences, Tehran 1416634793, Iran; 9Graduate School for Health Sciences, University of Bern, 3012 Bern, Switzerland; 10Department of Cardiology, University Hospital of Bern, University of Bern, 3012 Bern, Switzerland; 11Swiss Paraplegic Research, 6207 Nottwil, Switzerland

**Keywords:** buckwheat, cardiometabolic health, diet, cardiovascular diseases, fagopyrum

## Abstract

Buckwheat (BW) is suggested to have beneficial effects, but evidence on how it affects cardiometabolic health (CMH) is not yet established. We aimed to assess the effects of BW and/or its related bioactive compounds on cardiovascular disease (CVD) risk markers in adults. Five databases were searched for eligible studies. Observational prospective studies, nonrandomized or randomized trials were considered if they assessed BW, rutin or quercetin-3-glucoside intake and CVD risk markers. We adhered to the Preferred Reporting Items for Systematic Reviews and Meta-Analyses (PRISMA) guidelines for reporting. We selected 16 human studies based on 831 subjects with mild metabolic disturbances, such as hypercholesterolemia, diabetes and/or overweight. Eight studies, investigating primarily grain components, were included in the meta-analyses (*n* = 464). High study heterogeneity was present across most of our analyses. Weighted mean difference (WMD) for subjects receiving BW supplementation, compared to controls, were − 0.14 mmol/L (95% CI: −0.30; 0.02) for total cholesterol (TC), −0.03 mmol/L (95% CI: −0.22; 0.16) for LDL cholesterol, −0.14 kg (95% CI: −1.50; 1.22) for body weight, −0.04 mmol/L (95% CI: − 0.09;0.02) for HDL cholesterol, −0.02 mmol/L (95% CI: −0.15; 0.11) for triglycerides and −0.18 mmol/L (95% CI: −0.36; 0.003) for glucose. Most of the studies (66.7%) had concerns of risk of bias. Studies investigating other CVD markers were scarce and with inconsistent findings, where available. Evidence on how BW affects CMH is limited. However, the available literature indicates that BW supplementation in mild dyslipidaemia and type 2 diabetes may provide some benefit in lowering TC and glucose, albeit non-significant. Our work highlights the need for more rigorous trials, with better methodological rigor to clarify remaining uncertainties on potential effects of BW on CMH and its utility in clinical nutrition practice.

## 1. Introduction

Buckwheat (BW) (referring mainly to *Fagopyrum esculentum* and *F. tataricum*), a gluten-free pseudograin rich in fiber and bioactive compounds, has been suggested to positively affect cardiometabolic health [1]. The non-grain portion of the BW plant represents an important source of concentrated phenolics [2]. Globally, buckwheat demand has steadily increased [3], reflected in a rise in production to almost four million tons in 2021 alone [4]. Studies in animals have indicated that the intake of BW or BW-rich foods can influence glucose homeostasis and lipid metabolism, by modulating serum total cholesterol (TC) and triglycerides [5,6]. Such effects in the cardiovascular system have been partially attributed to the well-balanced amino acid composition of its proteins, dietary fiber content in the seeds and/or the presence of polyphenols such as rutin and quercetin-3-glucoside (hereinafter referred to as ‘quercetin’), that may confer protective properties against cardiovascular diseases (CVDs) [5,7]. Contrary to animal models, studies in humans have not yet established BW’s role as a dietary component for prevention of CVDs. Some human studies have indicated that BW can reduce serum lipid levels and blood pressure (BP), as well as improve body morphology parameters, while other studies have failed to show any favorable modification of CVD risk [5,6,7].

Despite the growing body of evidence and attention gained in cardiometabolic research in recent years, to the best of our knowledge, there is only one systematic review and meta-analysis on BW and its effects on CVD risk markers in humans [5], which has major methodological concerns that make interpretation of findings difficult. For example, data from different study designs, including cross-sectional with clinical trials, were pooled together without taking into account the correlation between pre- and post-intervention measures—something that can lead to seriously biased results and misleading conclusions [8]. Thus, it is not yet clear how and to what extent BW use and/or its bioactive compounds can influence CVD risk markers and exert cardiometabolic benefits.

Therefore, we systematically reviewed and meta-analyzed studies that investigated how dietary consumption or supplementation of BW, including bioactive compounds present in BW, was associated with a wide array of CVD risk markers, with the aim of understanding the association between BW intake and cardiometabolic health and translate its potential utility for clinical practice. Based on Population, Intervention, Comparison and Outcomes (PICO) criteria (see Table 1) we included only human adult subjects (≥18 years), exposed to a diet supplementation with buckwheat, rutin, quercetin and/or other related bioactives, compared to placebo, no buckwheat or other comparison. The outcomes we focused on were serum lipid profile, type 2 diabetes (T2D) and glucose homeostasis parameters, inflammatory markers, body morphology parameters, blood pressure, all-cause and CVD mortality, severity and/or clinical progression, markers of vasoconstriction/vasodilatation and/or markers of atherosclerosis, such as atherosclerotic plaque, arterial wall thickness, coronary artery calcification, intima media thickness, etc. With regards to study design, we considered all prospective cohort studies, case-cohort, nested-case control studies, randomized and non-randomized clinical trials.

## 2. Materials and Methods

### 2.1. Search Strategy and Study Selection

This work follows an established guide on conducting systematic reviews and meta-analyses for medical research [9], as well as PRISMA [10] guidelines for reporting. An experienced medical librarian systematically searched four electronic databases: EMBASE, MEDLINE (Ovid), Cochrane Central (Wiley) and Web of Science from inception until 17 January 2022 (date last searched); additionally, the first 200 results were downloaded from Google Scholar using *Publish or Perish* [11]. The following elements were used (1) cardiovascular risk and (2) buckwheat. The results were deduplicated using the *Bramer* method [12]. No study registries were searched, but Cochrane Central retrieves the contents of ClinicalTrials.gov and the World Health Organization’s International Clinical Trials Registry Platform. A detailed search strategy is outlined in the Appendix A (**‘*Search Strategy*’** section in the Appendix A). We additionally performed a hand search of the reference lists of included studies in the final analysis. Detailed inclusion and exclusion criteria can be found in the review protocol PROSPERO (ID: CRD42022307392).

#### 2.1.1. Inclusion Criteria

Studies were included if they (i) were conducted in humans (18+ years), (ii) were prospective cohort studies, case-control, case-cohort, nested-case control studies, randomized and non-randomized clinical trials and (iii) investigated the associations of buckwheat, rutin, quercetin and/or other BW-related bioactive supplementation with any of the following outcomes: serum lipid profile, T2D and glucose homeostasis parameters, inflammatory and oxidative stress markers, body morphology parameters, BP, all-cause and cardiovascular mortality, CVD severity and/or clinical progression, markers of vasoconstriction/vasodilatation and/or markers of atherosclerosis, such as atherosclerotic plaque, arterial wall thickness, coronary artery calcification, intima media thickness, etc.

No language restrictions applied.

#### 2.1.2. Exclusion Criteria

Nonhuman studies based on animal models or cell lines were excluded. We excluded studies with a cross-sectional design, reviews and meta-analyses, letters, conference posters/abstracts, editorials, case reports, book chapters and studies that did not specify the outcome. Human studies on subjects (<18 years) and pregnant women were also excluded from our analysis.

### 2.2. Screening, Data Extraction and Assessment of the Methodological Rigor of Included Studies

Two reviewers, who afterwards assessed the full texts of potentially eligible studies, independently evaluated titles and abstracts. Two reviewers also independently extracted the relevant information using a pre-defined data extraction form. Any disagreement between reviewers was settled by reaching a consensus or by consulting a third reviewer.

Study characteristics that were extracted included, but were not limited to, geographic location, study design, percentage of female subjects, size of control and intervention arm in the case of interventional studies, type of BW exposure/ intervention /control (i.e., placebo vs. active control), duration of intervention, participants’ health status, etc.

If necessary, a software (*Pixelruler*^®^ [13], Version: 10.0.0, Michael Rosenbaum, Ratzeburg, Germany ) was used to convert graphic data and/or authors were contacted via email. The methodological rigor of included studies was assessed by two independent reviewers using Cochrane Collaboration’s Tool Risk of Bias 2 (RoB2) [14] in the case of RCTs and ROBINS-I for non-randomized intervention studies [15].

RoB2 assesses five possible sources of bias, while ROBINS-I uses a similar system as RoB2, but with specification of the target trial and effect of interest, using signaling questions to inform judgements on risk of bias and assessments within potential bias domains. Information on the assessment of methodological rigor of studies and risk of bias are provided in (Appendix A).

### 2.3. Statistical Analysis and Evidence Synthesis

For clinical trials, intervention effects were defined as the pre-post differences in outcomes between BW-intervention and control arms at the end of the RCT. To calculate mean difference pre-post within one arm, a correlation of 0.8 was used to account for the correlation between measures [8]. All outcomes were continuous; therefore, mean differences (intervention minus control) of the treatment effects in CVD risk markers were presented as summary outcome measures. For data reported as medians, ranges or 95% confidence intervals (CI), means and standard deviations were converted as described elsewhere [16]. Random-effect models were used to obtain estimates of weighted mean differences (WMDs) and 95% CIs and fixed effects models were used as sensitivity analysis. For RCTs with crossover design, we used the data from the first study period only.

Study characteristics such as study location, duration, proportion of female participants, health status of study subjects and publication year, were used for assessment of heterogeneity with stratified analyses and random-effects meta-regression if eight or more studies were included in the meta-analysis, which were only performed if this criterion was met. For meta-analyses that included five or more studies, we explored the influence of any individual study on the pooled results by excluding one study at a time (see leave-one-out analysis in Appendix A).

Publication bias was evaluated through visual inspection of funnel plot and Egger’s test. All statistical analyses were conducted using STATA, release 16 (Stata Corp, College Station, TX, USA). RCTs that could not be quantitatively pooled (e.g., no control group, missing information/data or <3 studies with reporting markers of interest) and single-arm studies were qualitatively summarized.

## 3. Results

### 3.1. Included Studies

Of 3044 records yielded from the search strategy, 43 relevant full-text articles were retrieved, of which 10 studies met our eligibility criteria (Table 1). We screened the reference lists of those ten studies and identified an additional twenty-six records, of which six met our criteria, and thus including sixteen articles, based on sixteen interventional trials (three single-armed), for final analysis.

No observational studies were found to meet our criteria. The included studies involved eight-hundred and thirty-one participants and among them, only eight could be included in meta-analyses (*n* = 464 participants) (Figure 1). Six studies were conducted in Europe [17,18,19,20,21], eight in Asia-Pacific [22,23,24,25,26,27,28,29], one in Africa (Egypt) [30] and one in North America (Canada) [31]. Summarized results for the meta-analyses are provided in Appendix A.

For the eight studies included in the meta-analysis, sample size ranged from eight to one-hundred and sixty-five individuals (median twenty-eight subjects, interquartile range (IQR): 19–64) and the duration of the interventions from 1 to 15 weeks (median: 6 weeks, IQR: 4–9). Most studies (*n* = 12, 75%) included individuals with some form of metabolic disturbance (i.e., T2D, hypercholesterolemia, hyperlipidemia), while only four studies were exclusively conducted among healthy individuals. The majority of the studies (*n* = 11, 68.8%) investigated BW-based bread, biscuits, crackers or pasta/noodles, two reported on BW flour products and two investigated BW mixtures/herbs (Table 2). None of the studies took energy intake differences between trial arms into account.

Information on the methodological rigor assessment of the included studies and risk of bias is provided in Appendix A. Out of the four single-arm intervention studies, 75% were evaluated as having moderate risk of bias, mostly due to issues linked to confounding, selection of participants, and deviations from intended interventions. Among the twelve RCTs (eight of which were included in the meta-analysis), two trials (16.7%) had high risk of bias, six (50%) had some concerns of bias and four studies (33.3%) had low risk of bias.

### 3.2. Qualitative Synthesis

The scarce number of studies and the presence of high heterogeneity in control arms across studies did not permit a meta-analysis on BW and inflammation, oxidative stress markers, BP, some body morphology parameters (e.g., waist size, body mass index (BMI)), markers of vasoconstriction/vasodilatation and/or of atherosclerosis. There were no studies conducted, observational or otherwise, on the effect(s) of BW and/or its related bioactive compounds on all-cause and CVD-specific mortality, severity and/or clinical progression. A summary of these results is available in Appendix A.

#### 3.2.1. Inflammation Markers

We could not pool results for inflammatory markers reporting on BW effect on inflammatory markers, due to limited number of studies (*n* < 3) and lack of control arm.

In spite of these limitations, we provide a synthesis of the available studies. With regards to inflammatory and oxidative stress markers, Dinu et al. showed in a 2017 trial, that BW-enriched products produced no significant or clinically meaningful effect on a wide array of inflammatory markers [17].

This study examined interferon gamma, interleukin (IL) 4, 6, 8, 10, 12, IL-1 receptor antagonist and tumor necrosis factor (TNFα) among patients with non-celiac gluten sensitivity compared to a usual gluten free diet.

Previously, a trial of the same research group, involving 21 adults at high risk of CVDs, showed similar results on these markers [18]. Some additional markers of inflammation were included in this trial, i.e., macrophage inflammatory protein-1 alpha, malondialdehyde and reactive oxygen species from granulocytes and lymphocytes, as well as antioxidant capacity.

Only malondialdehyde (recognized as a marker of oxidative stress) and total antioxidant capacity levels were significantly affected, respectively, decreasing and increasing. Both trials found significant decreases in monocyte chemoattractant protein-1 (MCP-1) but were relatively small (*n* < 15) and compliance with intervention was unclear.

Further, from the included studies, two reported on C-reactive protein (CRP), but only one found a significant decrease [19,32]. One RCT reported no effect of BW-based, rutin-enriched cookies on high sensitivity CRP levels [21]. The same trial reported serum eosinophil cationic protein—a marker correlating to inflammation in the airways—together with myeloperoxidase and secretory phospholipase A_2_ group IIA (an independent risk marker for CVDs), were not affected by the BW treatment.

Mišan et al. showed a significant increase in levels of adiponectin in a three-arm crossover trial [32]. In this study, serum adiponectin levels were higher when the diet was enriched with BW-enriched instant porridge compared to a maize instant porridge. However, adiponectin levels declined during the crossover, despite the fact that weight and fat mass were similar between interventional periods in all three dietary interventions.

#### 3.2.2. Metabolic and Body Morphology Markers

Results for metabolic markers for which there was a sufficient number of studies (*n* ≥ 3), a control group or available data were pooled and meta-analyzed in further section of this report. However, for those markers that could not be pooled due to the aforementioned reasons, we present a narrative summary in this section.

Nishimura et al. reported levels of oxidized LDL cholesterol, thiobarbituric acid reactive substances (marker of lipid peroxidation that may reflect oxidative stress states) changed at 8 weeks, but there with no effect on its levels at 12 weeks, i.e., the end of BW intervention [25]. In this study, BMI and body fatness (BF) did not appear to be affected by the BW intervention. Three additional studies found similar null effects on BF and BMI [19,23,32].

Concerning BP, two studies reported a significant reduction in systolic BP [19,24], but two other studies found no significant changes in systolic BP after BW intervention [22,25]. Diastolic BP significantly decreased only in one study [19], but three other interventions reported no changes in DBP [22,24,25]. Three studies measured markers of atherosclerosis, of which two investigated vascular endothelial growth factor [17,18] and one reported atherogenic index [25]. In all these studies, there was no significant effect reported on any these markers after BW intervention.

Concerning glucose homeostasis, two studies found statistically significant reductions in glycated hemoglobin (HbA1c) after using non-grain components for the BW intervention [20,26]. Two other studies using grain components of *Tatary* BW did not find any effect on HbA1c [23,27]. It is uncertain how BW can influence HbA1c, as some of these studies lacked a control group and others had similar changes in HbA1c in the control group. In one the studies, BW intervention was not clearly described and was compared *head-to-head* with a drug among T2D patients [26].

Increased intake of whole-grain foods has been related to a reduced risk of developing diabetes and heart disease, with one underlying pathway for this relation attributed to increased insulin sensitivity. Insulin only significantly decreased in one intervention [23], but in two other studies, there was no significant or clinically relevant effect of BW intervention on insulin levels [27,33].

Similarly, the homeostatic model assessment for insulin resistance (HOMA-IR) was shown to significantly change in a small (*n* = 18), 8-week long RCT with BW-based pasta, bread, crackers and biscuits [33]. Another intervention with *Tartary* BW-enriched foods (e.g., kernel, noodle and powder) consumed *ad libitum*, among T2D patients, showed no notable changes in HOMA-IR, neither statistically nor clinically [23].

### 3.3. Meta-Analysis of Trials Assessing the Effect of Buckwheat Interventions on Lipid Profile

In total, eight studies (i.e., RCTs) contributed to the meta-analysis comparing the effects of BW interventions on lipid profile markers, i.e., total, LDL and HDL cholesterol, as well as TG. Concerning methodological quality, 50% of the studies had low risk of bias for TG, TC and HDL, but for LDL 57.1% had relatively low risk of bias. No evidence of publication bias was observed (Appendix A).

The weighted mean difference and 95% CIs for BW compared to control were −0.14 mmol/L; 95% CI: −0.30, 0.02 (8 trials, 464 participants in both arms, I^2^ = 76.5%, *p* = 0.001) for TC, −0.02 mmol/L; 95% CI: −0.15, 0.11 (8 trials, 464 participants, I^2^ = 73.9%, *p* < 0.001) for TG, −0.04 mmol/L; 95% CI: −0.09, 0.04 (8 trials, 464 participants, I^2^ = 56.2%, *p* = 0.025) for HDL-cholesterol and −0.02 mmol/L; 95% CI: −0.22, 0.16 (7 trials, 404 participants, I^2^ = 81.6%, *p* < 0.001) for LDL cholesterol. Results are shown in (Figure 2, Figure 3, Figure 4 and Figure 5 and Appendix A).

Stratified analyses showed (Appendix A) that for TC, WMD was significant in the case of studies conducted in Europe (WMD = −0.28 mmol/L, 95% CI: −0.37;−0.187, I^2^ = 0%, *p* = 0.57), including more than 50% female participants (WMD = −0.23 mmol/L, 95% CI: −0.41; −0.05, I^2^ = 59.6%, *p* = 0.04) or conducted among unhealthy participants (WMD = −0.28 mmol/L, 95% CI: −0.37; −0.19, I^2^ = 69.1%, *p* = 0.002), while null association was observed among the strata of other study characteristics. Most of the interventions (5/8) had mean value of TC out of the normal range at baseline in the intervention arm (Appendix A). For TG, WMD was consistent across all subgroup analyses, while one single study involving both healthy and unhealthy subjects (i.e., mix) showed higher levels with BW intake (WMD = 0.19 mmol/L, 95% CI: 0.03; 0.35).

TGs were significantly lowered among studies conducted before the year 2000 (WMD = −0.28 mmol/L, 95% CI: −0.43; −0.13, I^2^ = 0%, *p* = 0.53), while non-significant associations were observed in the other subgroup analyses. Only two studies out of eight had mean values of TGs out of the normal range at baseline in the intervention arm (Appendix A). HDL cholesterol decrease was associated with trial duration of more than 5 weeks (WMD = −0.09 mmol/L, 95% CI: −0.14; −0.04, I^2^ = 6.4%, *p* = 0.37); no difference from main results was observed among the strata of other study characteristics.

However, meta-regression analysis did not show any of the study characteristics to be a source of heterogeneity (*p*-value > 0.05) (Appendix A). At baseline, most of the studies (5/8) had mean values of HDL cholesterol out of the normal range in the intervention arm (Appendix A).

All studies were based primarily on interventions using BW flour, except the study by *Archimowicz-Cyrylowska* et al. (1996), which used non-grain component of BW in the form of herbal supplementation. In a posteriori analysis, we excluded this study from the meta-analyses, but this did not materially affect our results on blood lipids (Appendix A). Only a few studies (2/7) had mean values of LDL cholesterol at baseline out of the normal range in the intervention arm (Appendix A). *Leave-one-out* sensitivity analysis in general showed consistent results with the main findings, except for the meta-analysis on TC and HDL-C for which removing the study by Stringer et al. (WMD = −0.20 mmol/L, 95% CI: −0.33; −0.07) and Mišan et al. 2017 (WMD = −0.05 mmol/L, 95% CI: −0.10; −0.03,), respectively, showed reductions in TC and HDL-C with use of BW (Appendix A).

### 3.4. Meta-Analysis of Trials Assessing the Effect of Buckwheat Interventions on Body Weight and Glucose

Data on body weight were reported in three randomized controlled trials representing 343 participants in both arms, all of included studies judged of high methodological quality. Figure 6 shows the pooled results from the random-effects model combining WMD for the impact of BW intake on body weight. The results show no significant effect in the BW intervention arm in comparison with controls (WMD = −0.14 mmol/L; 95% CI: −1.50, −1.22, I^2^ = 0.0%, *p* = 0.990). No evidence of publication bias was observed (Appendix A).

Data on fasting blood glucose concentrations were reported in six randomized controlled trials representing a total 312 participants in both arms, based on the results of the meta-analysis. Half of the studies had a relatively low risk of bias. Figure 7 shows the pooled results from the random-effects model combing the WMD for the effect of BW interventions on fasting glucose concentration among study participants. Results show a borderline reduction in fasting blood glucose concentration after buckwheat intervention in comparison with to control arms (WMD = −0.18 mmol/L; 95% CI: −0.36, 0.00, I^2^ = 76.5%, *p* = 0.001).

A posteriori analysis—excluding the study by Archimowicz-Cyrylowska et al. (1996)—that used a non-grain component of BW in the form of herbal supplementation showed a significant reduction for glucose after BW supplementation (WMD = −0.19 mmol/L, 95% CI: −0.37; −0.01) (Appendix A). In the leave-one-out sensitivity analysis showed that removing the study by Mišan et al. 2017 from the meta-analysis, BW intervention was associated with a significant reduction in blood glucose levels (WMD = −0.30 mmol/L, 95% CI: −0.38; −0.22), while the results did not differ when removing the other studies (Appendix A). Overall, high heterogeneity was present in our meta-analyses, except for the meta-analysis on body weight.

## 4. Discussion

In this systematic review and meta-analysis, dietary supplementation with BW foods was not consistently associated with CVD risk markers and the results suggest a modest beneficial effect of BW interventions on TC and glucose levels, albeit non-significant. Poor methodological rigor of available studies, mainly concerning limited sets of CVD risk markers investigated on small samples, with no prospective studies available, as well as use of different types of BW (e.g., leaves, flour, cookies, etc.) limits the current understanding of the role of BW on cardiometabolic health and its utility to maximize metabolic benefits. Heterogeneity in the meta-analyses part was high, suggesting caution in the interpretation of our results.

Of note, our results contradict findings from a previous meta-analysis showing that BW interventions were consistently associated with improvements in serum lipid profile [5]. This can be partially explained methodology, as the previous meta-analysis pooled results from distinct study designs (i.e., cross-sectional, randomized, etc.), while using a heterogeneity test to define use of fixed or random effects models.

Evidence on exposure–outcome relationships can be inferred from many types of studies, including RCTs, cohort studies, case-control studies, cross-sectional analyses, ecological studies and animal studies. Each study type has characteristic strengths and weaknesses. For example, RCTs are the most robust method for dealing with confounding, but they are often conducted with strict inclusion and exclusion criteria, meaning that trial participants may be unlikely to be fully representative of the general population, as well as being carried out over relatively short durations.

Case-control studies are well suited for understanding the risks linked to rare outcomes, but they may be subject to recall bias for past exposure. Animal studies are widely used in evaluating the risks of consumer products and environmental risks but may not be generalizable to humans. Study design and analysis impacts causal interpretation and understanding of the results.

In our work, we took this fact into account and applied a stricter methodological rigor, differentiating study designs, as well as conducting a more structured evaluation of BW and cardiometabolic health. To the best of our knowledge, this review contains the largest number of human intervention studies to date, i.e., with three additional studies in the meta-analysis compared to the aforementioned review. Furthermore, it important to consider within-subject correlation in pre-post analysis, as failure to do so can lead to meta-analysis with misleading statistical results and inherent biases. To address this. we have used a coefficient of 0.8 in the calculations for the standard deviation of the mean difference [8].

Moreover, there are other reports that have attempted to ascertain the potential health benefits of consuming BW as a food, supplement, remedy or possible pharmaceutical agent. Recent work has focused on BW’s role in health and disease, especially investigating effects on lipid profile, BP, glucose and body weight, but the majority of these claimed effects in the literature are based on data extrapolated from in vitro studies or animal models [6,7].

Although animal models are vital in understanding physiological mechanisms and elucidating the potential health relevance, human intervention studies are scarce and inconsistent in supporting BW benefits identified in nonhuman studies. In the literature, a cross-sectional study with a questionnaire-based assessment of oat and buckwheat intakes showed a significant reduction in both systolic (−3.1 mm Hg, *p* < 0.001) and diastolic (−1.3 mm Hg, *p* < 0.01) BP [34]. Due to the study design (i.e., cross-sectional), this survey was not included in our analysis. Nevertheless, it is worth mentioning that the authors highlighted that water-soluble fiber but not total dietary fiber was independently associated with BP. Hence, a possibility exists that there is an effect of BW on these parameters, since BW has higher levels of soluble than insoluble fiber. Only two cross-sectional studies [34,35] have suggested that consumption of BW seeds may be a preventative factor for hypertension, dyslipidemia and hyperglycemia. However, the inherent limitations of a cross-sectional design render these findings indicative.

There are some results on the role of *Tatary* BW in human nutrition, showing that foods made from grain components of *Tartary* BW may have preventive effects against several chronic diseases, including obesity, CVDs, gallstone formation and hypertension [36]. However, these findings are almost entirely based on in vitro and nonhuman models, with little relevance to human health. Additionally, such effects are hypothetically attributed to resistant starch, protein, and phenolic substances in the grain, and to the interactions among these constituents, without any methodological high-quality study design in humans supporting these claims. Of note, in the systematic review part of our report, several studies used *Tatary* BW as intervention [21,22,23,25,26,27].

Results were not consistent with regards to effect of *Tatary* BW on CVD markers and study designs were very heterogeneous, with little, if any, considerations on compliance with the BW intervention, including no control arm or head-to-head comparison drugs. In our review, a consistent finding that aligns with previous reviews on BW [2,37], is the fact that many of the studies on BW were published before 2010, indicating an additional complexity with regards to methodological and technical procedures used. This aspect warrants further exploration, but it is a reminder that additional caution should be exercised when interpreting the literature on BW with considerations of separation, extraction, formulation, and processing methods.

Another observation is that BW is commonly a basis for noodle recipes in Asia. However, in Europe, BW flour is used in pancakes and crepes, as a common ingredient in gluten-free products, to which coeliac patients are particularly exposed. With such a wide use of BW at population level, it is reasonable to assume that the borderline effect of BW interventions on total and HDL cholesterol warrants further investigation, regardless of the questionable methodological rigor of evidence so far. According to BW’s degree of processing and food matrix, the primary mechanism of action may differ, but we speculate that this mechanism may include slower gastric emptying, the inhibition of hepatic cholesterol synthesis and/or enhanced fecal excretion of cholesterol and bile salts (see Figure 8).

Similar effects have been observed from dietary fibers in general [38] and it is not possible to differentiate the effect of quercetin or rutin from the effects of fibers present in BW. Nevertheless, the gel-forming attributes of soluble fibers in BW may be a basis for the borderline effect on some of lipids and glucose [39,40,41]. It is not clear whether different types of BW supplementation (e.g., grain vs. non-grain components) may have physiologically different effects on health. Herbal supplementation from non-grain components could be a richer source of natural phytochemicals, but this area of BW research remains speculative and to be elucidated in the future.

Processing can also influence the BW matrix and its composition. It may be the case that a single compound (i.e., rutin or quercetin) can influence several physiological functions, but also several BW compounds may affect a single defined physiological mechanism.

Processing, such as the type that disrupts the food matrix, may indirectly influence digestibility and/or bioavailability of BW nutrients, but can also degrade functionality by altering the structure of its components (e.g., depolymerisation of rutin or quercetin, lipid coalescence or protein denaturation) and/or their interaction. This warrants further research to confirm the specific effects and the mechanisms involved.

Over the past five decades, there have been efforts to document and establish health benefits from BW [5,6,7,36]. Despite its potential to improve human health, BW remains understudied in nutrition and clinical settings. Although the bioactive components present in BW are implicated in beneficial human health effects, future studies should focus on how insufficiently studied BW phytocompounds (such as phenolic acids and polyphenols) are metabolized in humans and influence cardiometabolic health. Thus, findings from this systematic review and meta-analysis might help guide future research to explore the potential health-promoting components of BW, which in turn will shed light on any health benefit this crop may deliver and its potential to be incorporated into human diet for optimal health.

Although our report is the largest review of RCTs and human interventional studies to date, concerns about the scarcity of studies, heterogeneity and methodological rigor concerns undermine establishment of causal inferences. The available interventional studies on BW have multiple limitations and flaws regarding sample size, intervention/follow-up time, confounders, blinding, randomization and allocation issues, which reinforce the need for more and better trials on the topic. In spite of the sensitivity analyses we performed to address limitations of the available evidence base, caution should be paid in interpreting some results as pooled studies had differing health characteristics. This constrains strong casual inferences and generalizability of findings, but we believe it can further stimulate the exploration of BW phytochemicals and their role(s) in human health. In addition, for crossover designs we used data collected from the first period only. Future studies should explore further interactions of BW bioactive components with health (Figure 8).

## 5. Conclusions

In conclusion, this review included randomized and non-randomized trials focusing on the BW supplementation on CVD risk markers. Overall, the magnitude of the associations between BW supplementation intervention and CVD risk markers was small and inconsistent. Given the distinction between exposures and type of BW component, subgroup analysis indicated that BW supplementation in mild dyslipidemia and T2D may provide some benefit in lowering TC and glucose, albeit non-significant. Modest positive associations were found in meta-analyses for weight and glucose homeostasis. In concert, findings from this systematic review and meta-analysis are of low certainty due to the poor methodological rigor and presence of heterogeneity across studies. However, given the increasing consumption of BW, understanding the effect its grain and non-grain components have on CVD markers is important for improving evidence-based recommendations in improving and/or maintaining good cardiometabolic health. Future trials should focus on methodological rigor and explore BW leaves and other non-grain components, which can be richer in bioactive compounds [2].

## Figures and Tables

**Figure 1 jpm-12-01940-f001:**
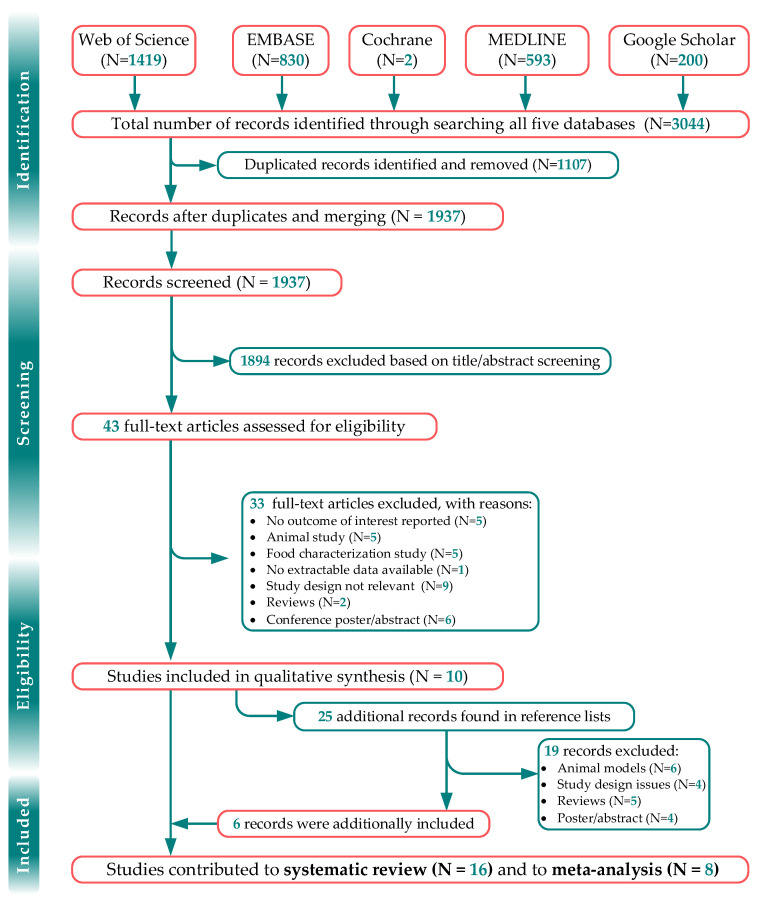
PRISMA flow.

**Figure 2 jpm-12-01940-f002:**
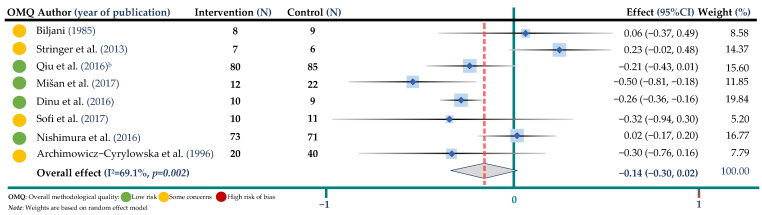
Meta-analysis on buckwheat supplementation and total cholesterol. ^b^ indicates the Qiu study according to Table 2 [17,18,20,22,25,28,31,32].

**Figure 3 jpm-12-01940-f003:**
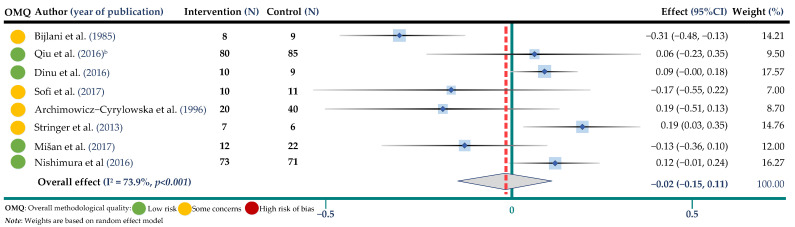
Meta-analysis of buckwheat supplementation and triglycerides. ^b^ indicates the Qiu study according to Table 2 [17,18,20,22,25,28,31,32].

**Figure 4 jpm-12-01940-f004:**
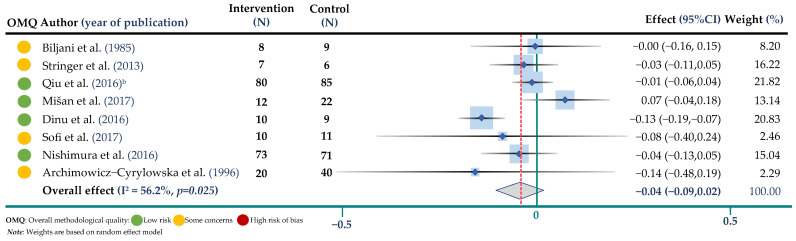
Meta-analysis of buckwheat supplementation and HDL cholesterol. ^b^ indicates the Qiu study according to Table 2 [17,18,20,22,25,28,31,32].

**Figure 5 jpm-12-01940-f005:**
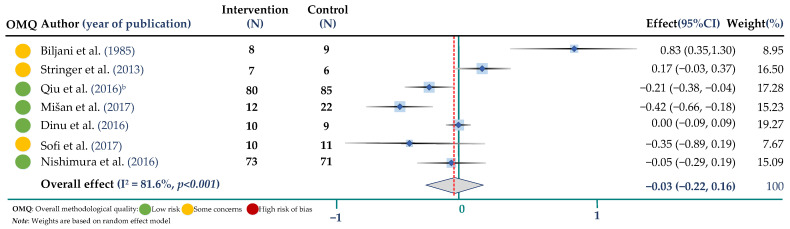
Meta-analysis of buckwheat supplementation and LDL cholesterol. ^b^ indicates the Qiu study according to Table 2 [17,18,22,25,28,31,32].

**Figure 6 jpm-12-01940-f006:**
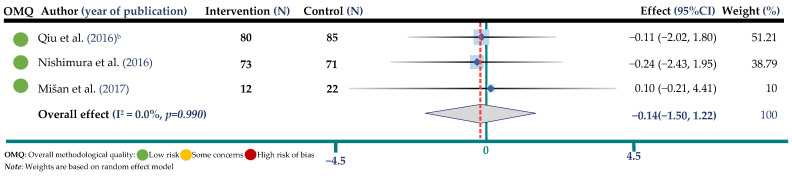
Meta-analysis of buckwheat supplementation and body weight. ^b^ indicates the Qiu study according to Table 2 [22,25,32].

**Figure 7 jpm-12-01940-f007:**
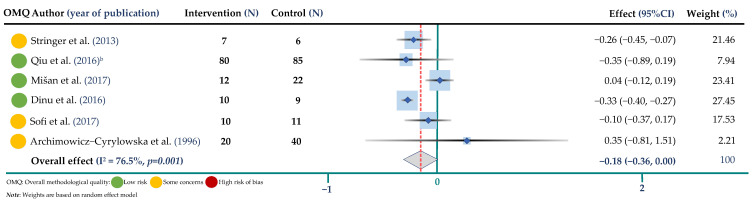
Meta-analysis of buckwheat supplementation and glucose. ^b^ indicates the Qiu study according to Table 2 [17,18,20,22,23,31].

**Figure 8 jpm-12-01940-f008:**
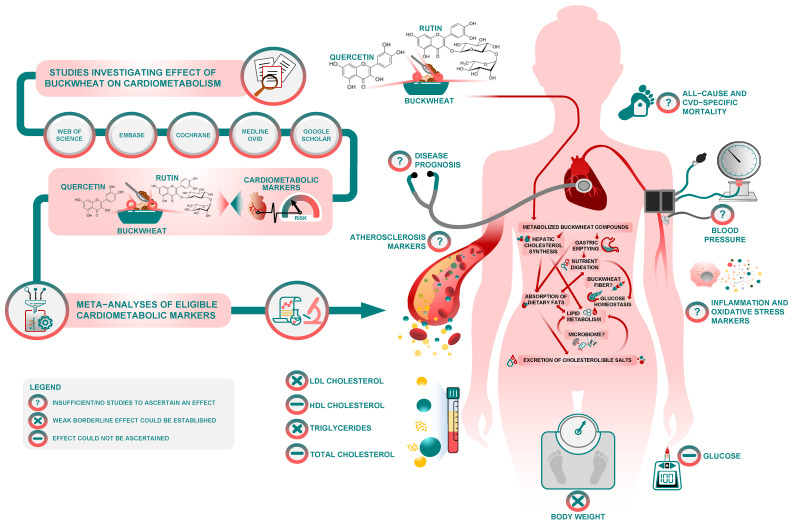
Visual summary, mechanisms and potential avenues for future research.

**Table 1 jpm-12-01940-t001:** PICOS criteria for inclusion of studies.

Parameter	Criterion
**Population**	Human adults (≥18 years)
**Interventions/exposures**	Diet supplementation with buckwheat, rutin, quercetin and/or other buckwheat related bioactives
**Comparisons**	Placebo, no buckwheat or other comparison
**Outcomes**	Serum lipid profile, type 2 diabetes and glucose homeostasis parameters, inflammatory and oxidative stress markers, body morphology parameters, blood pressure, all-cause and cardiovascular mortality, cardiovascular disease severity and/or clinical progression, markers of vasoconstriction/vasodilatation and/or markers of atherosclerosis, such as atherosclerotic plaque, arterial wall thickness, coronary artery calcification, intima media thickness, etc.
**Study design**	Prospective cohort studies, case-cohort, nested-case control studies, randomized and non-randomized clinical trials

**Table 2 jpm-12-01940-t002:** Characteristics of studies included in systematic review.

No.	Study (Year)	Location	Study Design	Sample (N)	Females (%)	Study Population	Mean Age (SD) *	Duration (Weeks)	Washout Information	Intervention	Control	Primary Aim of the Study	Risk of Bias **	Ref.
1	Bijlani et al. (1985)	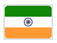 India	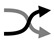	8	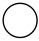 0%	Healthy	18–34	12	2 wks WO	Wholegrain BW flatbread	Breakfast cereals	Lipid profile and glucose tolerance	Some concerns	[26]
2	Bijlani et al. (1984)	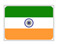 India	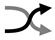	12	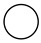 0%	Healthy	18–22	4	No WO	Sieved BW flatbread	Staple lunch cereal	Lipid profile and glucose tolerance	High	[27]
3	Stringer et al. (2013)	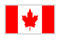 Canada	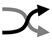	13	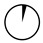 15%	Healthy/T2D	49 (11.5)	1	No WO	Wholegrain BW crackers	Rice cracker	Food intake and glucose homeostatsis	Some concerns	[29]
4	Dinu et al. (2017)	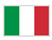 Italy	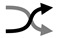	19	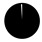 95%	NCGS	45.3 (10)	6	No WO	BW-enriched pasta, tacks, biscuits and flakes	Gluten-free diet	Gastrointestinal health	Low	[15]
5	Zheng et al. (1991)	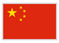 China	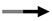	19	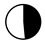 47%	NIDDM	53.8 (29–64)	12	NA	Tartary BW flour	No control	Lipid profile	Moderate risk	[25]
6	Stokić et al. (2015)	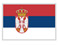 Serbia	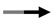	20	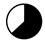 65%	At risk of CVDs	59.5 (12.6)	4	NA	Wholegrain BW-enriched wheat bread	Wheat bread	Lipid profile	Moderate risk	[17]
7	Shakib et al. (2011)	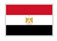 Egypt	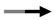	20	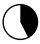 40%	HC and NIDDM	R: 30–50	6	NA	BW-yogurt mixture	No control	Lipid profile and glucose homeostatsis	Serious risk	[28]
8	Sofi et al. (2016)	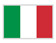 Italy	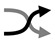	21	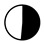 52%	At risk of CVDs	51.3 (13.4)	8	8 wks WO	BW-based bread, pasta, biscuits and crackers	Wheat-based bread, pasta, biscuits and crackers	CVD risk markers	Some concerns	[16]
9	Mišan et al. (2017)	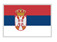 Serbia	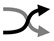	34	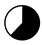 62%	HC	46 (8.2)	5	3 wks WO	BW-enriched instant porridge	Maize instant porridge	Lipid profile and inflamation	Low	[30]
10	Archimowicz-Cyrylowska et al. (1996)	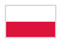 Poland	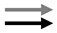	60	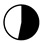 53%	T2D	20–75	12	NA	BW herb mixture	*Troxerutin* or *Ruscus* mixture	Retinopathy and lipid profile	Some concerns	[18]
11	Zhao and Guan (2003)	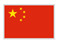 China	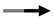	60	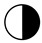 50%	T2D	R: 26–67	8	NA	BW flour	No control	CVD risk profile	Moderate risk	[22]
12	Wieslander et al. (2011)	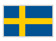 Sweden	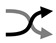	62	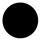 100%	Healthy	46 (10)	4	2 wks WO	Tatary BW cookies	Common BW cookies	Lipid profile and inflamation	High risk	[19]
13	Huang et al. (2009)	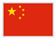 China	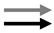	70	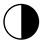 50%	T2D	53 (8.2)	8	NA	BW mixture	Control drug	Diabetic nephropathy	Some concerns	[24]
14	Qiu et al. (2016a)	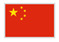 China	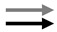	104	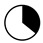 39%	T2D	58.8 (9.4)	4	NA	Tartary BW foods	Diet plan and nutritional education	Lipid profile and glucose homeostatsis	Low	[20]
15	Nishimura et al. (2016)	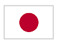 Japan	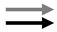	144	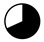 70%	Healthy	54.1 (8.9)	12	NA	Rutin-rich Tartary BW noodles and cookies	Wheat-based noodles and cookies	Antioxidant effects	Some concerns	[23]
16	Qiu et al. (2016b)	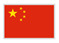 China	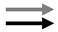	165	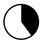 41%	T2D	56.9 (10.4)	4	NA	Tartary BW-rich foods	Refined cereals (i.e., rice or wheat flour)	Renal function	Low	[21]

* Buckwheat interventions indicate common buckwheat, unless otherwise specified. ** For in-depth study methodological assessment see Appendix A. Values are provided as mean age and standard deviation, unless otherwise indicated. WO: Washout period; BW: buckwheat; HC: subjects with hypercholesterolemia; T2D: type 2 diabetes; NCGS: Non-celiac gluten sensitivity; NIDDM: Non-insulin-dependent diabetes mellitus. Studies are ordered by sample size, from very small (red) to very large (green) 
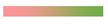
—based on the sample size range of included studies. 
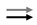
 Parallel RCT study design; 
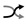
 crossover RCT study design; 
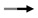
 dietary intervention study (e.g., before-after intervention, etc.); NA: not applicable.

## Data Availability

Not applicable.

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
