# Peer review of "Buckwheat and Cardiometabolic Health: A Systematic Review and Meta-Analysis"

_jpm, 2022, doi:10.3390/jpm12121940_

Round 1

Reviewer 1 Report

Review presents a very systematic comprehension of studies on cardiometabolic benefits of buckwheat. Text is well written and flows very well. The review would be very useful for researchers designing future studies as highlighted in the review  There are some spelling mistakes here and there which can be improved.

Author Response

We are pleased with the Reviewer’s interest towards our paper and we thank the Reviewer for acknowledging the importance of our systematic review and meta-analyses. In line with the Reviewer’s observation we have further checked for all minor spelling typos and corrected them accordingly. We share the same opinion with the Reviewer with regards to the potential utility of this work and believe that it would be a great resource for researchers designing future studies or working in the field.

Reviewer 2 Report

The paper is a systematic review and meta-analysis that aimed to underline the association between Buckwheat intake and cardio metabolic health. The authors adhered to the PRISMA guidelines during the conduction of the systematic review. However some changes are still needed

·      In the abstract it should be clearly mentioned in adherence to the PRISMA guidelines

·      The PICO statement should be clearly mentioned at the end of the introduction section

·      The I2 were nearly in all the meta-analysis major than 50, indicating a important heterogeneity among the included studies. Therefore the results should interpreted with caution and no firm conclusions can be done. This should be clearly mentioned in the Result and Discussion sections. For this reason all interpretations and conclusions in the entire manuscript should be softened due to this important statistical limitation. 

Author Response

Response to Reviewer 2

We thank the reviewer for his comments and for reviewing of our work. We have revised our Abstract and added a new sentence to indicate our adherence to PRISMA guidelines as follows (lines 29-30):

We adhered to the Preferred Reporting Items for Systematic Reviews and Meta-Analyses (PRIS-MA) guidelines for reporting”

In addition, we have included a statement for the PICO, in the revised version of the manuscript, at the end of the Introduction part (as advised by the Reviewer) and we refer the reader to the PICO table as well, as follows (lines 79-89):

Based on Population, Intervention, Comparison and Outcomes (PICO) criteria (see Table 1) we included only human adult subjects (≥18 years), exposed to a diet supplementation with buckwheat, rutin, quercetin and/or other related bioactives, compared to placebo, no buckwheat or other comparison. The outcomes we focused on were serum lipid profile, type 2 diabetes (T2D) and glucose homeostasis parameters, inflammatory markers, body morphology parameters, blood pressure, all-cause and CVD mortality, severity and/or clinical progression, markers of vasoconstriction/vasodilatation and/or markers of atherosclerosis, such as atherosclerotic plaque, arterial wall thickness, coronary artery calcification, intima media thickness, etc. With regards to study design we considered all prospective cohort studies, case-cohort, nested-case control studies, randomized and non-randomized clinical trials.”

Eventually, we agree with the reviewer’s recommendation for further softening the language due to the presence of heterogeneity in our meta-analyses. In line with the Reviewer’s suggestion, we have included several statements, in addition to existing ones, to indicate this limitation to the reader. This language has been added to the revised version of the manuscript as follows:

  • In the abstract (lines 32-33): “High study heterogeneity was present across most of our analyses.”
  • In the Results section (lines 382-383): “Overall, high heterogeneity was present in our meta-analyses, except for the meta-analysis on body weight.”
  • In the Discussion (lines 396-397): “Heterogeneity in the meta-analyses part was high, suggesting caution in interpretation of our results.”
  • In the Conclusion part we have re-written the existing statement with an emphasis to heterogeneity (lines521-524): “In concert, findings from this systematic review and meta-analysis are of low certainty due to the poor methodological rigor and presence of heterogeneity across studies.”

Round 2

Reviewer 2 Report

Non